# Applications of Hydrogels in Osteoarthritis Treatment

**DOI:** 10.3390/biomedicines12040923

**Published:** 2024-04-22

**Authors:** Xin Gan, Xiaohui Wang, Yiwan Huang, Guanghao Li, Hao Kang

**Affiliations:** 1Department of Orthopedics, Tongji Hospital, Tongji Medical College, Huazhong University of Science and Technology, Wuhan 430030, China; d202282263@hust.edu.cn; 2The Center for Biomedical Research, Department of Respiratory and Critical Care Medicine, NHC Key Laboratory of Respiratory Diseases, Tongji Hospital, Tongji Medical College, Huazhong University of Science and Technology, Wuhan 430030, China; d202282145@hust.edu.cn; 3School of Materials and Chemical Engineering, Hubei University of Technology, Wuhan 430068, China; yiwanhuang@hbut.edu.cn

**Keywords:** Hydrogel, Biomaterials, Osteoarthritis, Interdisciplinary therapy, Drug delivery

## Abstract

This review critically evaluates advancements in multifunctional hydrogels, particularly focusing on their applications in osteoarthritis (OA) therapy. As research evolves from traditional natural materials, there is a significant shift towards synthetic and composite hydrogels, known for their superior mechanical properties and enhanced biodegradability. This review spotlights novel applications such as injectable hydrogels, microneedle technology, and responsive hydrogels, which have revolutionized OA treatment through targeted and efficient therapeutic delivery. Moreover, it discusses innovative hydrogel materials, including protein-based and superlubricating hydrogels, for their potential to reduce joint friction and inflammation. The integration of bioactive compounds within hydrogels to augment therapeutic efficacy is also examined. Furthermore, the review anticipates continued technological advancements and a deeper understanding of hydrogel-based OA therapies. It emphasizes the potential of hydrogels to provide tailored, minimally invasive treatments, thus highlighting their critical role in advancing the dynamic field of biomaterial science for OA management.

## 1. Introduction

Osteoarthritis (OA) represents a group of severe degenerative diseases profoundly affecting human health. It is estimated that OA impacts approximately 500 million people worldwide, accounting for a prevalence of 7%. Notably, the rate escalates to 30% among elderly populations and postmenopausal women [1,2]. OA predominantly targets knee and hip joints, with pathological features including articular cartilage degeneration, subchondral bone remodeling, osteophyte formation, and synovitis [3,4]. The pathogenesis of OA encompasses a range of biological processes, such as genetics, metabolism, biomechanics, and immunity (Figure 1A). These factors interact synergistically, contributing to the disease’s complexity [5,6,7,8,9,10,11]. In OA pathogenesis, the roles of cartilage, subchondral bone, synovium, infrapatellar fat pad and menisci are significant [3,12]. A crucial aspect is the disruption of the extracellular matrix (ECM) of chondrocytes [13]. There is a decrease in the synthesis of Type II collagen (Collagen II) and proteoglycan (Aggrecan), coupled with increased production of matrix-degrading enzymes like matrix metalloproteinases (MMPs) and A Disintegrin and Metalloproteinase with Thrombospondin Motifs (ADAMTS). This results in the breakdown of ECM and consequently destruction of cartilage [14,15]. Additionally, macrophages in the synovium and subchondral bone secrete inflammatory mediators such as tumor necrosis factor-alpha (TNF-α) and interleukin-1β (IL-1β). These mediators further amplify the destruction of the chondrocyte ECM and induce a marked increase in mitochondrial aerobic metabolism in chondrocytes. The heightened metabolism leads to excessive production of ROS that damage chondrocytes, ultimately triggering chondrocyte apoptosis and autophagy [16,17].

OA affects multiple tissues in joints, including the cartilage, subchondral bone, synovium, infrapatellar fat pad, and meniscus. OA leads to pain and functional impairment through cartilage degeneration, subchondral bone changes, synovial inflammation, and meniscal damage, hence a comprehensive treatment strategy is required. Consequently, a holistic treatment approach, termed “treating the joint as a whole,” has been advocated for systemic OA management. Subchondral bone supports cartilage, absorbs shock and provides joint stability. The synovium produces synovial fluid to lubricate and nourish the joint. It is reported that the infrapatellar fat pad and the synovium share the same anatomical functions [18,19]. Synovial cells in IPF can secrete APOE and MDK to regulate the senescence of chondrocytes, and this is confirmed by single-cell RNA sequencing in latest findings [20,21]. The meniscus stabilizes the knee joint, distributes load, reduces friction [22]. Inflammation of the synovium can exacerbate osteoarthritis symptoms. Damage to the infrapatellar fat pad, changes in subchondral bone, and meniscal injuries can all promote degenerative changes in the joint, accelerating the progression of osteoarthritis [23,24]. This implies that OA treatment should include the surrounding tissues besides cartilage, forming an integrated treatment approach [25,26]. However, current primary OA treatments, both pharmacological and surgical, are limited in their ability to reverse cartilage damage. Addressing cartilage repair and regeneration remains a significant clinical challenge, highlighting the urgent need for novel therapeutic strategies [2,27].

Hydrogels, a class of hydrophilic, three-dimensional network gels, generally consist of polymer frameworks and water. They serve as ideal carriers for direct drug or cell therapy delivery to joints [28,29]. Hydrogels enable the sustained release of drugs and bioactive substances, offering long-term therapeutic effects and aiding in the repair of damaged cartilage areas [30]. Additionally, they provide support to damaged cartilage, reducing stress on the affected area and decelerating OA progression [31,32,33]. The development of injectable hydrogels represents a minimally invasive treatment option, greatly reducing patient discomfort during the treatment process (Figure 1B). The introduction of hydrogel microneedles further refines drug delivery, facilitating non-invasive treatment [34,35,36]. Recent research has been investigating the incorporation of bioactive molecules and nanoparticles into hydrogels to augment their therapeutic efficacy [32,37,38]. Moreover, the emergence of protein-based hydrogels, with their suitable mechanical properties and excellent biological characteristics, marks a novel approach in OA treatment [39,40]. The concept of “superlubricating” hydrogels has been introduced to innovatively treat OA by reducing cartilage inflammation through exceptionally low friction [41,42]. Additionally, the novel material Aggregation-Induced Emission (AIE) substantially improves the responsiveness of hydrogels [43]. Hydrogels, traditional materials with a long history in materials science, continue to exhibit immense potential in OA treatment, propelled by continuous innovations.

Recent years have witnessed a marked increase in hydrogel research. Data from the Science Citation Index (SCI) database reveals that the total number of hydrogel-related research publications reached 96,864 in the past decade. Notably, 34% of these publications occurred in the last three years, indicating a consistent upward trend. Specifically, 509 articles (excluding reviews) have focused on hydrogel applications in OA treatment (Figure 2A), and 1,905 articles have been dedicated to cartilage repair, showing annual growth in this research domain [6]. Hydrogels derived from natural materials, aparticularly those based on hyaluronic acid and alginate, are prevalent in OA treatment, largely due to their superior biocompatibility and bioactivity [44]. Furthermore, the convenience and adaptability of injectable hydrogels have made them a significant choice in OA therapy (Figure 2B). Conversely, synthetic hydrogels, despite their unique advantages like customizable physicochemical properties, have seen limited practical use. These research developments not only underscore the relevance of hydrogel technology in biomedicine but also highlight its potential for future advancements in this field [45]. With ongoing progress in material science and bioengineering, hydrogels are poised to play a pivotal role in OA treatment and other medical applications.

This article offers a comprehensive review of hydrogel applications in OA treatment. The primary focus in OA treatment is on enhancing chondrocyte function, a significant departure from approaches centered on cartilage regeneration. The review begins by categorizing hydrogels used in OA therapy, discussing different cross-linking methods in hydrogel fabrication. It then provides an overview of the incorporation of bioactive substances in hydrogels, shedding light on how these substances enhance therapeutic effectiveness. Subsequently, the review explores recent advances in various new types of hydrogels within the OA field, examining them in terms of new materials, treatment techniques, and fundamental principles. It also covers the latest innovations in hydrogel technology pertinent to OA therapy. Moreover, the article anticipates future research trajectories for hydrogels, offering valuable insights for developing novel hydrogel formulations. This article highlights the novel hydrogels in OA therapy including hydrogel microneedles, hydrogel microspheres, multiple responsive hydrogels, decellularized matrix hydrogel, protein hydrogels and superlubricating hydrogels, distinguished with the former articles, and plays a key role in inspiring the creation of innovative hydrogels and suggesting new therapeutic strategies for OA, highlighting the dynamic nature of this field and its significant potential impact on OA treatment.

## 2. The Classification of Hydrogels Used in the Treatment of OA

Hydrogels can be classified based on their material origin into natural and synthetic hydrogels (Figure 3). Regarding their physical structure, they are categorized into homogeneous and composite hydrogels. In terms of cross-linking methods, they are divided into physically and chemically cross-linked hydrogels. Additionally, based on responsiveness to stimuli, they are differentiated into responsive and non-responsive hydrogels. In practical applications, hydrogels are predominantly used in their composite form, which may not align precisely with the aforementioned classifications. For instance, Gelatin Methacryloyl (GelMA) combines natural gelatin components with synthetic methacrylate, thus discussions here are restricted to whether the raw materials of synthetic hydrogels are naturally derived or artificially synthesized. Different types of hydrogels exhibit distinct mechanical properties, which significantly influence their clinical applications in OA therapy [46]. Synthetic hydrogels often boast enhanced strength and durability, making them suitable for load-bearing applications, whereas natural hydrogels offer better biocompatibility and bioactivity, ideal for promoting tissue regeneration. Composite hydrogels combine these advantages, providing both mechanical robustness and biological functionality, thereby supporting cartilage repair and mitigating OA progression [29].

### 2.1. Natural Hydrogels

#### 2.1.1. Hyaluronic Acid

Hyaluronic Acid (HA), the main component of the extracellular matrix (ECM) in human cartilage tissues and joint fluid, has garnered increasing attention in OA research. HA not only provides vital lubrication to joints but also plays an essential role in sustaining joint health and cartilage function [47]. It is pivotal in regulating the regeneration, proliferation, metabolism, and apoptosis of chondrocytes, and stimulates ECM synthesis in chondrocytes during inflammatory responses [48]. Recent advancements in HA modification and cross-linking, along with its incorporation into microspheres, 3D printing, and responsive hydrogels, have broadened its applications in OA therapy [49]. For example, enhancing HA hydrogels with biocompatible polymers such as Polyethylene Glycol (PEG) increases their stability. Additionally, forming Methacrylated HA (HAMA) by combining HA with methacrylic acid yields a temperature-sensitive, injectable material suitable for diverse OA treatments [50].

#### 2.1.2. Collagen

Collagen, a primary constituent of cartilage tissue and a crucial element of ECM hydrogels in cartilage, is integral to collagen-based hydrogel development. These hydrogels, which emulate the structure and function of natural cartilage, create a conducive environment for repairing and regenerating damaged joints. Not only are collagen-based hydrogels biocompatible and biodegradable, but they also facilitate chondrocyte adhesion, growth, and differentiation, thus aiding the joint cartilage repair process [51]. A notable advantage of collagen-based hydrogels is their capacity to preserve joint lubrication and minimize friction in knee joints [52]. This makes collagen-based hydrogels a promising avenue in hydrogel technology research.

#### 2.1.3. Alginate

Alginate, a natural polysaccharide sourced from seaweed, forms hydrogels by reacting with polyvalent cations, such as calcium ions (Ca^2+^), to create a stable three-dimensional network structure. This structure closely resembles the ECM of cartilage tissue, offering an optimal microenvironment for chondrocyte growth and cartilage repair [53]. Alginate hydrogels are particularly versatile and innovative in their ability to cross-link with various ions. Calcium ions are the most common choice for cross-linking alginate, resulting in a stable gel network that serves as an effective drug delivery carrier, tissue engineering scaffold, or wound dressing [54,55,56,57]. Alginate hydrogels cross-linked with copper ions (Cu^2+^), known for their wound healing and antibacterial properties, are applied in wound care where antimicrobial action and accelerated healing are essential [58,59]. Iron ions (Fe^2+^/Fe^3+^), utilized in alginate hydrogels, bestow paramagnetic properties, suitable for creating magnetically responsive hydrogels [60,61]. Zinc ions (Zn^2+^), when used to cross-link alginate hydrogels, enhance cell proliferation and differentiation [62,63]. Alginate hydrogels cross-linked with multivalent ions like aluminum (Al^3+^) or chromium (Cr^3+^) exhibit amphoteric chemical properties, enabling pH-responsive behavior [64,65]. Alginate hydrogels continue to be a central focus in OA treatment research.

#### 2.1.4. Chitosan

Chitosan, a natural polysaccharide extracted from crustacean exoskeletons, primarily consists of N-acetyl-D-glucosamine units. Its distinctive polysaccharide structure facilitates the formation of a three-dimensional networked gel in water, providing an excellent hydration environment [66]. Chitosan’s strong nucleophilic characteristics enable the addition of functional side chains through electrophilic reactions. It can be processed into hydrogels using both physical and chemical cross-linking methods. Owing to its outstanding biocompatibility, biodegradability, non-toxicity, and gel-forming properties, chitosan hydrogels are widely employed in OA treatment [67].

#### 2.1.5. Gelatin

Gelatin, a natural protein derived from the partial hydrolysis of collagen in animal skin, bones, and connective tissues, contains the amino acid sequence arginine-glycine-aspartic acid. In water, gelatin forms a stable three-dimensional network structure akin to the ECM of natural cartilage tissues. This structure promotes the adhesion and growth of chondrocytes, thus facilitating the repair of damaged cartilage tissue. When cross-linked with methacrylic acid, gelatin forms GelMA hydrogels, notable for their photosensitivity and injectability. Post-injection, GelMA can solidify into a gel within the body under UV irradiation. It can also be pre-formed into microspheres using microfluidic technology before injection, establishing it as a versatile and widely used multifunctional hydrogel [68]. Due to their cost-effectiveness and exceptional biological properties, gelatin hydrogels are extensively researched in OA treatment.

### 2.2. Synthetic Hydrogels

Synthetic hydrogels are typically produced through the chemical synthesis of organic polymers. They often exhibit superior mechanical properties and more precisely controllable chemical and physical characteristics compared to natural hydrogels. Examples of synthetic hydrogels include Polyvinyl Alcohol (PVA), Polyacrylic Acid (PAA), Poly(N-isopropylacrylamide) (PNIPAM), and Polyethylene Glycol (PEG). These synthetic hydrogels can be cross-linked with other materials to enhance their functionality and performance in various applications.

#### 2.2.1. Polyvinyl Alcohol

Polyvinyl Alcohol (PVA) is distinguished by its excellent water solubility. PVA hydrogels, known for their biocompatibility and tunable mechanical properties, closely resemble the hydration and mechanical characteristics of soft tissues, making them ideal for replicating natural articular cartilage. These hydrogels are synthesized through physical cross-linking methods, such as freeze-thaw cycles, or chemical cross-linking to form stable three-dimensional network structures. In OA treatment, PVA hydrogels are predominantly utilized as artificial substitutes or repair materials for joint cartilage, providing effective cushioning and shock absorption that mitigate joint inflammation and pain [69]. However, PVA hydrogels exhibit relatively low bioactivity, and their mechanical strength and durability may be insufficient for long-term joint pressure and wear, particularly in knee applications. Their non-biodegradable nature also considerably restricts their use [70]. Therefore, current research is directed towards enhancing PVA hydrogels with suitable material technologies to better fulfill therapeutic requirements.

#### 2.2.2. Polyacrylic Acid

Polyacrylic Acid (PAA) is characterized by its high hydrophilicity, forming a three-dimensional network capable of significant water absorption. The mechanical strength and stability of PAA hydrogels are relatively robust, enabling them to emulate essential cartilage properties such as cushioning and elasticity. This renders PAA hydrogels suitable as artificial substitutes for joint cartilage. Nevertheless, the biocompatibility of PAA hydrogels is somewhat limited, potentially causing local tissue inflammatory responses. Furthermore, their pronounced hydrophilicity may lead to excessive water uptake, resulting in a volumetric increase that could impact their stability and function within the joint cavity. Moreover, the mechanical strength and durability of PAA hydrogels sometimes fall short, necessitating cross-linking with side chain groups to improve their performance [71,72].

#### 2.2.3. Poly(N-isopropylacrylamide)

Poly(N-isopropylacrylamide) (PNIPAM) hydrogels are extensively studied in OA treatment due to their unique temperature-sensitive properties. They exhibit a phase transition that allows them to switch from a solution state to a gel state below their lower critical solution temperature (LCST). This feature grants PNIPAM hydrogels excellent injectable properties [73]. This phase transition enables PNIPAM hydrogels to form a stable gel at body temperature, offering support and cushioning to damaged joints. They are convenient and cost-effective [74]. However, the biocompatibility and bioactivity of PNIPAM hydrogels are relatively limited, and their stability in the body requires improvement. Thus, modifications through cross-linking or blending with other polymers are essential to enhance their durability under the continuous mechanical stress of joints [74,75].

#### 2.2.4. Polyethylene Glycol

Polyethylene Glycol (PEG) is known for its high-water solubility, exceptional biocompatibility, and versatility. The molecular weight of PEG can be varied through synthetic techniques, permitting control over its physicochemical properties, including viscosity, solubility, and permeability. Additionally, PEG’s chemical structure can be covalently altered to introduce specific functional groups [76]. PEG-based materials are widely used in clinical settings. PEG hydrogels, by forming highly hydrated three-dimensional network structures, emulate the natural environment of joint cartilage. They provide support and protection to damaged cartilage, buffer joint stress, and are unlikely to cause immune responses or inflammation. Due to the tunability of their chemical structure, PEG hydrogels are also effective in drug delivery systems, enabling the controlled release of anti-inflammatory agents or growth factors [77,78,79,80]. However, the mechanical strength and stability of PEG hydrogels are somewhat inadequate for enduring joint stresses, necessitating modifications in practical applications to bolster their mechanical robustness.

#### 2.2.5. Polyphosphazene

Polyphosphazene, a synthetic polymer with a backbone of alternating nitrogen and phosphorus atoms, displays unique chemical and physical properties, including high adaptability, good biocompatibility, and biodegradability. Tailoring the polyphosphazene side chains via chemical modification can adjust its characteristics, such as water solubility, elasticity, and degradability, thus enhancing its applicability in medical materials [81]. Employed as a novel therapeutic carrier, polyphosphazene hydrogels have been used in wound repair, attributed to their excellent antibacterial properties. Researchers, including Ni et al., have utilized these hydrogels in drug delivery systems for the controlled release of anti-inflammatory drugs and growth factors [82,83]. However, as an emerging hydrogel material, research into its biosafety and effectiveness is still limited. Its mechanical strength remains somewhat lacking, necessitating more comprehensive studies to fully explore its potential applications.

## 3. Cross-Linking Methods of Hydrogels

Hydrogel synthesis is an interdisciplinary endeavor that combines chemistry, biology, and material science, striving to create polymer networks with specific functionalities and properties. Recognizing that natural hydrogels often exhibit lower mechanical performance and synthetic hydrogels may pose biocompatibility challenges, researchers are focusing on developing composite hydrogel systems. These systems merge the benefits of both natural and synthetic polymers, incorporating bioactive substances to more effectively address OA treatment. The cross-linking methods of hydrogels are primarily divided into physical and chemical cross-linking (Figure 4). Physical cross-linking typically involves reversible and non-covalent bonding structures, brought about by temperature changes or physical agents. Conversely, chemical cross-linking creates covalent bonds between polymer chains, yielding a more robust and enduring network structure. This method allows for enhanced control of the hydrogel’s properties, including mechanical strength, degradation rate, and drug release profiles.

### 3.1. Physical Cross-Linking

Physical cross-linking encompasses freeze-thaw cycles, ionic interactions, and self-assembling cross-linking. Self-assembling cross-linking depends on non-covalent interactions between molecules, such as hydrogen bonding, hydrophobic interactions, and electrostatic forces.

#### 3.1.1. Ionic Interaction

Ionic cross-linking hinges on the interactions between ions of opposite charges to create cross-linking points. Its advantages include simplicity, the absence of complex chemical reagents, and the capacity to proceed under mild conditions. This method employs multivalent cations (e.g., calcium and aluminum ions) to cross-link with multivalent anion polymers (like alginate) [84]. Huang used Zr^4+^ to cross-link polyampholyte to fabricate mechanically strengthened hydrogel with a Young’s modulus of 39.2 MPa and 3.7 MPa of tensile strength [85]. In this approach, oppositely charged ions attract, establishing stable connection points between polymer chains, which culminate in a three-dimensional network structure.

#### 3.1.2. Self-Assembling Cross-Linking

This method utilizes non-covalent interactions, including hydrogen bonding and hydrophobic interactions, which prompt molecular self-assembly into a stable hydrogel structure. Hydrogen bond cross-linking, a prevalent physical cross-linking approach in polymer and hydrogel materials, is contingent on hydrogen bond formation [86]. Hydrophobic interaction cross-linking in the hydrogel arises from hydrophobic groups, typically due to water molecule repulsion that facilitates the attraction of non-polar or weakly polar molecules. In the case of Polyethylene Glycol (PEG) hydrogels, PEG chains modified with hydrophobic groups, such as methyl and ethyl, aggregate in water to form stable cross-linking points [87].

#### 3.1.3. Freeze-Thaw Cycles

The freeze-thaw method, frequently utilized for preparing materials like Polyvinyl Alcohol (PVA), relies on physical changes induced by temperature variations without involving chemical reactions or requiring chemical cross-linking agents. During the freezing phase, water molecules within the hydrogel crystallize into ice, prompting polymer chains to aggregate and form dense regions. These regions, stabilized through physical entanglement and interactions such as hydrogen bonds, create cross-linking points. Upon thawing, as the ice melts, the polymer chains maintain the physically cross-linked structure formed during the freeze. Repeated freeze-thaw cycles further reinforce this structure, enhancing the hydrogel’s mechanical strength and stability. The microphase separation occurring in the freeze-thaw process facilitates the formation of hydrogels with distinctive porous structures and network morphologies [88,89].

### 3.2. Chemical Cross-Linking

Chemical cross-linking encompasses covalent cross-linking, radiation cross-linking, and enzyme-catalyzed cross-linking. It involves the creation of stable covalent bonds linking polymer chains via chemical reactions. Covalent cross-linking necessitates cross-linking agents, commonly including glutaraldehyde, sodium periodate, and isocyanates. Radiation cross-linking is initiated using ultraviolet light, gamma rays, or electron beam radiation. Enzyme-catalyzed cross-linking employs specific enzymes, like transglutaminase, to catalyze the cross-linking of proteins or peptides.

#### 3.2.1. Covalent Cross-Linking

Covalent cross-linking, a method that forms stable covalent bonds between polymer chains, typically involves reactive groups such as hydroxyl, amine, and carboxyl groups reacting with cross-linking agents like glutaraldehyde, sodium periodate, and isocyanates. The Schiff base reaction enables grafting functional groups onto hydrogels, creating multifunctional composite hydrogels [90,91]. Jung et al. utilized this reaction to develop oxidized alginate and gelatin hydrogels (COS-SA) with excellent biocompatibility and cartilage-protective effects [92]. These reactions create permanent connections between polymer chains, conferring lasting mechanical and chemical stability to the hydrogels. Hydrogels formed by covalent cross-linking are notably stable and can be tailored for specific applications.

#### 3.2.2. Light Cross-Linking

Light cross-linking uses light, typically ultraviolet, to initiate chemical reactions that cross-link polymer chains. This method involves polymers with photosensitive groups or the addition of photoinitiators, with common groups including epoxy and benzophenone derivatives. Upon exposure to specific light wavelengths, these groups activate and form covalent cross-links. Photocross-linking provides precise spatial and temporal control and can be executed without high temperatures or harmful solvents. This precision makes photocross-linking particularly apt for creating complex three-dimensional structures in applications like 3D printing and the fabrication of microspheres for precision medicine [93]. However, this method often produces cytotoxic oxygen free radicals.

#### 3.2.3. Enzyme-Catalyzed Cross-Linking

Enzyme-catalyzed cross-linking employs specific enzymes, such as transglutaminase, to catalyze reactions between groups (e.g., lysine and glutamic acid residues in peptide chains) to create stable covalent cross-links. This technique typically uses biocompatible natural polymers, like proteins or peptides, under mild conditions, thus preserving bioactive substances. Hydrogels cross-linked via enzyme catalysis are useful as carriers for bioactive substances, cell culture matrices, or adhesives in tissue engineering [94].

## 4. Hydrogel Drug Delivery Systems and Cell Carrier Therapies

In OA treatment, the intrinsic bioactivity of hydrogels is limited and often inadequate for alleviating cartilage damage independently. Hence, the use of hydrogels as carriers for drugs or bioactive substances is widespread. The application of drug-loaded nanoparticles in hydrogels enables targeted release and controlled drug release rates, improving therapeutic effects [95]. Additionally, hydrogels function as cell carriers in tissue engineering for OA treatment (Figure 5).

### 4.1. Drugs

A variety of drugs, including anti-inflammatory agents such as non-steroidal anti-inflammatory drugs (NSAIDs) and corticosteroids, along with cartilage repair-promoting drugs, are used in OA treatment [96]. Research often combines anti-inflammatory drugs with cartilage repair agents to more effectively address joint damage and support cartilage repair [97]. NSAIDs, particularly selective COX-2 inhibitors like celecoxib and rofecoxib, are commonly delivered through composite hydrogels. Their generally good water solubility enables direct mixing with hydrogels for sustained effectiveness [98,99]. Corticosteroids, such as hydrocortisone and prednisone, are also prevalent. Moreover, drugs like diacerein, an IL-1 receptor antagonist, demonstrate clinical efficacy, but their slightly poor water solubility may necessitate encapsulation in microspheres or liposomes before incorporation into hydrogels for prolonged release [100,101].

### 4.2. Bioactive Substances

The bioactive substances utilized in OA treatment predominantly include proteins, cytokines, biologics, and exosomes. Agents such as Transforming Growth Factor-beta (TGF-β) and Bone Morphogenetic Proteins (BMPs) have proven effective in promoting cartilage regeneration and repair [102,103]. Glucosamine and chondroitin sulfate can significantly alleviate joint cartilage degeneration. Small molecule compounds, like Kartogenin, encourage mesenchymal stem cells to differentiate into chondrocytes. Platelet-Rich Plasma (PRP) aids in reducing inflammation in cartilage and subchondral bone. Hydrogels incorporating these bioactive components have demonstrated substantial therapeutic efficacy in OA treatment [104,105,106,107,108,109,110]. For instance, Yuan et al. developed a temperature-sensitive, injectable hydrogel using hydroxypropyl chitosan (HPCH) loaded TGF-β, stromal cell-derived factor 1α (SDF-1α) as well as poly(lactic-co-glycolic) acid (PLGA) microspheres containing kartogenin (KGN). This hydrogel facilitated controlled drug release and exhibited cartilage-protective effects, offering significant clinical translation potential [111,112,113]. Additionally, hydrogels containing small interfering RNA and transcription-regulating viruses have been developed. J. Maihöfer et al. used an alginate hydrogel (IGF-I/AlgPH155) loaded with recombinant adeno-associated virus (rAAV) vectors encoding human Insulin-like Growth Factor I (IGF-I) in large animal models, showing substantial efficacy in alleviating knee OA [114].

Exosomes, tiny vesicles secreted by cells containing proteins, miRNA, and other bioactive components, exhibit anti-inflammatory, antioxidant, immune-modulating, and cell proliferation-enhancing properties. They represent a novel approach in OA treatment research. Zhang et al. formulated a hydrogel from alginate-dopamine, chondroitin sulfate, and regenerated silk fibroin (AD/CS/RSF) loaded with MSC-derived exosomes (EXO) [115]. Pang et al. developed GelMA-based hydrogels photo-crosslinked to encapsulate mesenchymal stem cell-derived nanovesicles (MSC-NVs) for animal injections [116]. Zeng et al. created a modified chitosan dual-drug hydrogel system loaded with mesenchymal stem cell-derived exosomes (MSC-exo) and icariin (ICA). Similarly, Yin et al. produced polyethylene glycol-hyaluronic acid hydrogel microspheres infused with mir-99a-3p modified adipose-derived stem cell exosomes [117]. These studies indicate that hydrogels enriched with exosomes effectively promote cartilage regeneration and mitigate joint inflammation, offering promising applications. However, challenges such as high production costs and limited availability hinder their widespread application, and further clinical validation is necessary.

### 4.3. Stem Cell Carrier Therapy

Hydrogels, with their ECM-like structure, excellent biocompatibility, and porosity, are ideal as cellular scaffolds in stem cell-based therapies for OA. Mesenchymal stem cells (MSCs) are frequently employed for cell loading, with adipose-derived stem cells also being widely used due to their abundant availability [118]. Yan et al. developed a DNA supramolecular hydrogel embedded with MSCs to reduce friction in knee joint cartilages [24]. Zhong et al. utilized extracellular matrix hydrogels loaded with bone marrow-derived MSCs for direct treatment of damaged menisci in OA [119]. Zhang et al. facilitated OA treatment by incorporating xenogeneic MSCs into HA hydrogels [120]. In cartilage defect repair, loaded cells actively contribute to cartilage regeneration at the defect site. In contrast, OA treatment primarily focuses on improving chondrocyte function rather than cartilage structure repair. Thus, compared to OA treatment, cell carrier therapy has broader applications in cartilage repair.

## 5. Advanced Applications of Hydrogels in OA Treatment

As hydrogel applications continue to advance, the sophistication of composite hydrogels is increasing, with widespread applications in treating conditions like OA, burns, ulcers, tumors, diabetes, ophthalmic disorders, and cardiovascular diseases. Hydrogels used as cellular scaffolds in tissue engineering have proven effective in repairing nerves and blood vessels. They show immense potential in facilitating tissue repair and enhancing drug delivery, offering promising avenues in various medical fields [121,122]. However, a primary limitation in their biological efficacy is the relatively low mechanical strength of hydrogels. Ongoing research and development are expected to yield new approaches for OA treatment, with innovations in hydrogel materials, drug delivery methods, processes, and structures significantly enhancing hydrogel functionality.

### 5.1. Hydrogel Microneedles

Hydrogel microneedle is a novel method in drug administration. As a drug delivery system, microneedle technology, typically involves creating fine, needle-like structures on the skin’s surface using materials with good biocompatibility and biodegradability such as hyaluronic acid and polyethylene glycol (Figure 6). 3D-printing is the most frequently used method to form microneedles. Combine the microneedle with proper hydrogel, it enables painless drug delivery [123]. In contrast to conventional hydrogel injections for OA treatment, hydrogel microneedles offer substantial advantages, including minimal invasiveness, reduced infection risk, and ease of use. Enhanced with nanotechnology, hydrogel microneedles can also enable more precise drug delivery and treatment monitoring. Zhang et al. developed polydopamine hydrogel microneedles that function similarly to traditional plasters, providing sustained release and prolonged effects, thereby enhancing patient quality of life during treatment [124]. Lin et al. applied hydrogels in traditional Chinese acupuncture, effectively penetrating rat and rabbit cartilage cell layers, managing inflammation in both superficial and deep cartilage, and reducing subchondral bone sclerosis [125]. These advancements underscore the potential of hydrogel microneedle technology in OA therapy. Its non-invasive nature is likely to improve patient quality of life and treatment adherence, marking it as a significant research area in OA therapy.

### 5.2. Hydrogel Microspheres

Hydrogel microspheres are created by processing hydrogel materials into tiny, spherical particles. These microspheres, notable for their high-water content and biocompatibility, are capable of systematically releasing drugs (Figure 7). Their applications span drug delivery and tissue engineering, making them a focal point in recent hydrogel research [127,128]. Common methods for preparing microspheres include emulsion techniques and microfluidic technology. Post-formation, hydrogels can be cross-linked with various functional groups to augment their functionalities. HA microspheres, for instance, facilitate sustained drug release, enhancing treatment efficacy and duration. Xia et al. developed dual-responsive hydrogel microspheres by anchoring methacrylated gelatin (GelMA) and phenylboronic acid (PBA) onto hyaluronic acid methacrylate (HAMA), loaded with dihydromyricetin (DMY). These microspheres activate the SIRT3 signaling pathway, preserving organelle balance in chondrocytes, and reducing autophagy and apoptosis [129]. Xiao et al. incorporated chemokines, macrophage antibodies, and engineered cell membrane vesicles (sEVs) into HAMA hydrogel microspheres, achieving precise macrophage reprogramming and regulation [130,131]. Zuo et al. synthesized highly permeable nanogel microspheres using triphenylphosphine (TPP) and HAMA, enhancing cartilage and subchondral bone metabolism via ROS scavenging [132]. Li et al. developed liposome-anchored teriparatide (PTH (1-34)) encapsulated in GelMA hydrogel microspheres, mitigating IL-1β-induced inflammation in ATDC5 cells by modulating the PI3K/AKT signaling pathway [133].

### 5.3. Responsive Hydrogels

Hydrogel microspheres can be tailored to modify their structure, offering a broad spectrum of functionalities. Owing to their large specific surface area, they provide more efficient drug release properties than traditional hydrogels and can more effectively achieve joint lubrication. Responsive hydrogels constitute a class of hydrogels that respond to specific environmental stimuli [134]. In the field of OA treatment, responsive hydrogels primarily encompass temperature-responsive, pH-responsive, enzyme-responsive, magnetic-responsive, ROS-responsive, and mechanical-responsive hydrogels (Figure 8A). Temperature-responsive hydrogels expand or contract at a critical temperature (LCST) and solidify under body temperature conditions (Figure 8B), providing localized treatment, a feature inherent in most injectable hydrogels [135]. Yi et al. developed a robust thermosensitive hydrogel by cross-linking collagen with PLGA and polyethylene glycol (PEG) [136]. Y. H. et al. synthesized a thermosensitive chitosan-gelatin hydrogel that, through delivering glutathione, mitigated mitochondrial dysfunction, thereby reducing chondrocyte apoptosis and autophagy [137]. HA is also frequently used in thermosensitive hydrogels; Zhou et al. formulated an injectable HAMA hydrogel microsphere responsive to hypoxia and MMP13 (Figure 8E) that degrades under MMP13 and releases drugs in hypoxic conditions [138]. Mechanical-responsive hydrogels react to pressure and friction (Figure 8B). Liu et al.’s mechanical stress-responsive hydrogel, created by cross-linking PAA with polyaniline (PANI), instantaneously hardens under pressure, providing support for stressed areas of cartilage during walking [139]. Wu et al.’s HSPC nanoliposome-encapsulated gelatin (HG) facilitates friction force-responsive drug release [140]. In OA, cartilage cells under inflammation generate significant amounts of ROS; Scognamiglio et al.’s ROS-responsive hydrogel, engineered with borate-crosslinked lactose-modified chitosan (CTL), releases drugs in the presence of ROS (Figure 8G), exhibiting excellent biocompatibility and antioxidant properties [141]. Jiang et al. incorporated magnetic responsiveness into GelMA-HAMA hydrogels using boronized neodymium iron boron (NdFeB) (Figure 8F), enhancing the simulation of pressure on OA cartilage and establishing a groundwork for future research [142]. AIE materials, distinguished from traditional luminescent materials, shine brightly in aggregated or solid states. Employing AIE, exceptionally sensitive and biocompatible pH-responsive (Figure 8D), temperature-responsive, and enzyme-responsive hydrogels can be engineered [143,144,145]. The responsiveness of hydrogels enables precision treatment. Various responses contribute to hydrogel formation, drug release, and degradation. This lays the foundation for the precision and diversification of hydrogel functions and signifies a key future research direction in hydrogel development [122].

### 5.4. Novel Hydrogel Materials

The sources of hydrogels have expanded beyond conventional natural substances and synthetically produced polymers, with emerging materials markedly improving their functionality. The decellularized matrix, obtained from natural tissues by eliminating cellular elements, consists of extracellular matrix components. This matrix maintains the structure and bioactivity of the original tissues, offering a conducive environment for cell adhesion, proliferation, and differentiation [119,146,147,148]. As a result, decellularized matrices are increasingly utilized in the fabrication of biological scaffolds, aiding in the repair and regeneration of damaged tissues (Figure 9). Hydrogels, in contrast, provide adjustable physical characteristics and drug delivery capacities, thereby enhancing the matrices’ functionality. Yuan et al. crafted a hydrogel from human mesenchymal stem cell extracellular matrix that demonstrated significant efficacy in mitigating cartilage damage in the meniscus of immunodeficient mice, markedly improving cartilage repair in the affected areas [149]. Similarly, Qiang et al. employed the extracellular matrix of porcine menisci chondrocytes, combined with polyethylene glycol diacrylate (PEGDA), to synthesize a decellularized matrix hydrogel that successfully treated OA in rat knee joints [150]. Integrating hydrogels with ECM materials to formulate composites with outstanding physicochemical and biological properties emerges as a promising strategy in OA therapy [151,152].

Protein hydrogels are a distinct class of hydrogels, distinct from earlier variants. They consist of either purely synthetic proteins or combinations with other polymers to establish composite network structures. These hydrogels demonstrate customizable biochemical and mechanical properties, with notable degradability and biocompatibility. Fu and colleagues devised protein sequences using ferredoxin-like protein (FL) as a foundation. Individual monomers assemble into octamers via dityrosine bonds. In polar solutions, these FL octamers ((FL)_8_) remain in a folded state. Upon chemical denaturation, (FL)_8_ transforms into chain-like structures. In concentrated solutions, the interweaving of these protein chains results in robust cross-linking, a widely used physical cross-linking technique in polymer hydrogel preparation. This method depends on the physical entanglement and interaction of polymer chains, contrasting with the covalent bonds in traditional chemical cross-linking. Furthermore, denatured (FL)_8_ in dilute solutions reverts to its folded form under hydrogen bonding, forming collapsed structures. The refolded N-DC ((FL)_8_) exhibits enhanced physical properties and is transformed into a protein hydrogel resembling cartilage (Figure 10). This hydrogel exhibits exceptional mechanical properties, melding high hardness and toughness with biocompatibility. It has demonstrated promising therapeutic effects in treating OA in rats. The development of this hydrogel represents a significant advancement in OA treatment, heralding the use of artificially designed sequence protein hydrogels in this field. It provides valuable insights for future research and applications of hydrogels [40].

### 5.5. Superlubricating Hydrogels

Superlubrication is typically defined as a friction coefficient lower than 0.01, also termed ‘ultra-low friction’. Superlubricating hydrogels attain this extremely low friction coefficient through innovative chemical and physical designs, eclipsing the lubricative capabilities of conventional hydrogel materials (Figure 11). These hydrogels often replicate natural lubrication mechanisms, especially those in cartilage tissue. Primarily consisting of polymer networks, they can imbibe significant water quantities, forming stable, water-rich gels [153]. Incorporating specific lubricating molecules into the hydrogel, such as hyaluronic acid or polyethylene glycol, drastically reduces the surface friction coefficient. Lin and colleagues, mimicking the boundary lubrication of liposomes in joint cartilage, developed liposome-integrated hydrogel microspheres with a friction coefficient of about 0.01 [154]. Yang, drawing on the concept of bearing lubrication, synthesized porous amphiphilic ionic microspheres capable of releasing platelet-derived growth factor BB (PDGF-BB), markedly reducing knee joint friction upon intra-articular injection [155]. Inspired by the lubricative properties of ice, Zhao and team employed microfluidics to generate 2-methacryloyloxyethyl phosphorylcholine (MPC) modified hydroxyethyl methacrylate (HEMA) microspheres loaded with diclofenac sodium, achieving superlubrication and prolonged drug release in knee joints [156]. Han and associates utilized DMA and MPC to create branch crosslinked GelMA hydrogel microspheres, effectively reducing joint friction upon intra-articular administration, thus serving as joint lubricants [157]. These hydrogels act as joint lubricants, diminishing friction and wear on joint surfaces, thereby mitigating pain and retarding disease progression.

## 6. Prospective and Outlook

OA, a prevalent degenerative disease in the elderly, severely compromises their quality of life. The incidence of OA rises significantly with age and is more common in women than in men. This discrepancy is often linked to reduced estrogen levels during menopause, which contributes to osteoporosis, while genetics, trauma, and prolonged use are also key factors in the development of knee OA [1,2,158]. Treating joint damage poses a considerable clinical challenge: Cartilage, being terminally differentiated, exhibits limited self-renewal capacity and is challenging to regenerate once damaged. Additionally, alterations in the force lines in the affected area further hasten the degeneration of joint cartilage [3,159,160]. Existing conventional OA treatments, including pharmacotherapy and surgery, fail to cure the condition. In this context, hydrogels, as established biomaterials, have attracted significant interest in OA research. Their excellent biocompatibility, resemblance to cartilage cell extracellular matrix, adjustable properties, drug delivery capabilities, and functional versatility render them a compelling option in OA therapy. The ability of hydrogels to simulate the natural tissue environment and their potential in targeted drug delivery and tissue engineering highlight their importance in evolving OA treatment strategies. Current research and development in this field indicate a future where hydrogels may play a crucial role in effectively managing and potentially reversing OA’s effects.

Many studies have highlighted the potential of advanced hydrogels in treating OA, yet only a limited number of these materials have progressed to clinical trials. This gap primarily stems from the challenges associated with scaling laboratory findings to practical, clinical applications. However, there has been some success in using HA as a cell carrier in clinical settings. HA-based treatments for OA have demonstrated high biocompatibility, evidenced by a range of studies [161,162,163]. One notable example of an HA-based hydrogel that has received FDA approval for commercial use is HYMOVIS®. This hydrogel represents a significant advancement in OA treatment options available to patients, showcasing the effectiveness of such materials in clinical applications. Currently, more hydrogels are undergoing clinical studies to explore their potential for wider clinical use [164,165]. These emerging technologies are well-positioned for OA treatment, tackling the current challenges in hydrogel molding. Microfluidics technology enables precise control over hydrogel particle size. The integration of nanotechnology is expected to enhance the efficiency and specificity of hydrogel drug delivery at the microscopic level. The adoption of microneedle technology could revolutionize conventional drug delivery methods, improving safety and compliance in post-optimization treatments, thus facilitating painless OA treatment. Responsive hydrogels offer more accurate treatment aligned with disease conditions. Novel protein hydrogels are providing novel insights into treating cartilage inflammation [166]. Superlubricating hydrogels, focused on reducing friction between cartilages, have demonstrated significant efficacy in alleviating OA symptoms, particularly pain [145]. These developments have substantially enriched the hydrogel research field, injecting new energy and directions. With these advancements, hydrogels are increasingly recognized as key components of innovative medical solutions, especially in addressing the complexities and challenges of OA treatment.

As technological progress and medical needs advance, the limited mechanical strength of traditional hydrogels has proven inadequate for clinical applications. This necessitated the exploration and development of new hydrogel types, yielding several innovations recently. Mikyung and colleagues have introduced an injectable tissue interface prosthesis (IT-IC) with immediate gelation properties, utilizing multiple cross-linking methods. The IT-IC hydrogel incorporates biphenyl bonds and coordination bonds formed from conductive gold nanoparticles (AuNPs) interacting with PB groups in situ through biphenyl reduction, providing modest electrical conductivity suitable for various tissue repairs [167]. Advanced technologies like 3D printing and microfluidics have further amplified hydrogel efficacy. The use of 3D printing enables hydrogel structure customization to individual patient requirements, enhancing treatment personalization and precision. Zhang has developed a sono-ink that facilitates self-reinforcing cross-linking in hydrogels. Utilizing this, the deep-penetration acoustic volumetric printing (DAVP) technique was created, differing markedly from traditional 3D printing and not relying on photosensitive materials, making a wider range of hydrogels printable [168]. Ni and colleagues have ingeniously produced hydrogels with memory and delayed recovery features through 3D printing, capable of shape-responsive action and timed reversion to their original state [169].

Recent advancements in hydrogel research and application have markedly diversified their sources and enhanced their physical properties. This development has revolutionized traditional hydrogels. In OA treatment, hydrogels must fulfill criteria for biocompatibility, mechanical strength, biodegradability, and therapeutic convenience, elevating standards in the field. Traditional hydrogels, such as hyaluronic acid (HA), previously exhibited inferior mechanical properties, which have been greatly enhanced through modifications. Current research primarily focuses on discovering new hydrogel materials and improving existing ones. Prospects involve a deeper comprehension of hydrogel theory and ongoing advancements in materials science, anticipating more effective and precise hydrogel-based OA treatments. This promises improved options for OA patients and marks a significant evolution in biomaterial science. In summary, hydrogels, as versatile and adaptable biomaterials, are poised to play a crucial role in OA treatment. Their potential for personalized medicine, increased therapeutic efficacy, and minimally invasive application places them at the forefront of innovative treatments, promising to revolutionize OA management.

## 7. Conclusions

This review comprehensively examines hydrogels in OA therapy. The development of hydrogel applications in osteoarthritis treatment has progressed significantly over several decades. Initially, in the 1980s, research focused on natural hydrogels like hyaluronic acid and collagen, prized for their biocompatibility. By the late 1990s and early 2000s, synthetic hydrogels such as PVA and PEG emerged, offering enhanced mechanical properties and customization. The mid-2000s marked a pivotal advancement with the introduction of injectable hydrogel systems, minimizing invasiveness while maximizing therapeutic efficacy. The 2010s saw the development of multi-responsive hydrogels, capable of reacting to various biological stimuli like pH and temperature changes, which enabled more precise drug delivery. Most recently, the 2020s have integrated 3D printing with hydrogel fabrication, facilitating personalized treatment approaches by tailoring hydrogels to individual patient needs, thus optimizing therapeutic outcomes in osteoarthritis management. The advancements in hydrogel technology have significantly enriched treatment strategies for osteoarthritis, providing innovative and valuable approaches that hold considerable promise for enhancing therapeutic outcomes.

## Figures and Tables

**Figure 1 biomedicines-12-00923-f001:**
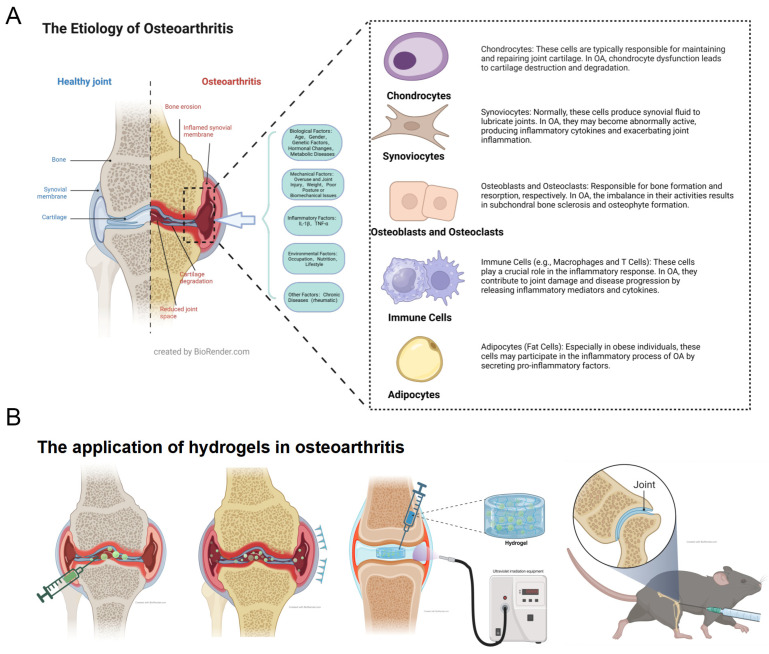
(**A**) Schematic Diagram of the Pathogenesis of OA and (**B**) Various Hydrogel Treatments for OA.

**Figure 2 biomedicines-12-00923-f002:**
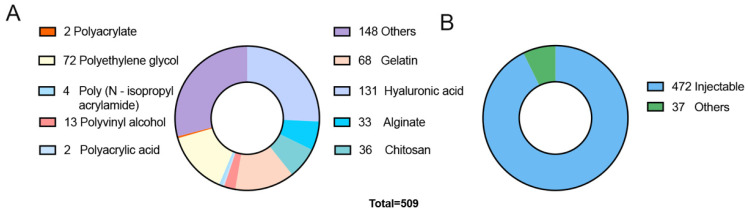
(**A**) Number of SCI indexed publications on various types of source-based hydrogels used in OA treatment. (**B**) Number of SCI indexed publications comparing injectable hydrogels to other hydrogel-based treatment methods.

**Figure 3 biomedicines-12-00923-f003:**
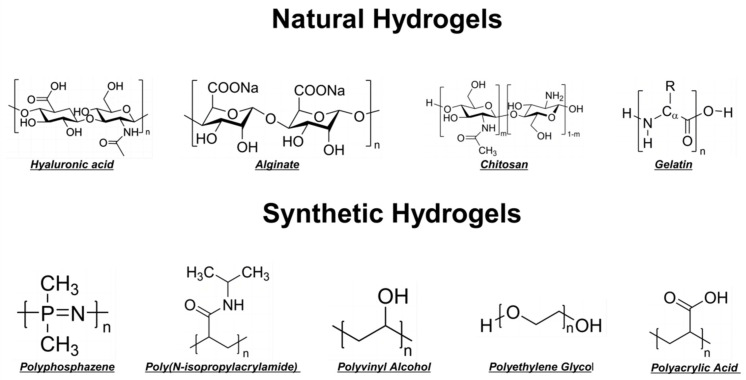
Classification and Structural Diagram of Hydrogel Raw Material Sources. Natural hydrogels include hyaluronic acid, alginate, chitosan, and gelatin. Synthetic hydrogels encompass poly(N-isopropylacrylamide), polyvinyl alcohol, polyethylene glycol, and polyacrylic acid hydrogels.

**Figure 4 biomedicines-12-00923-f004:**
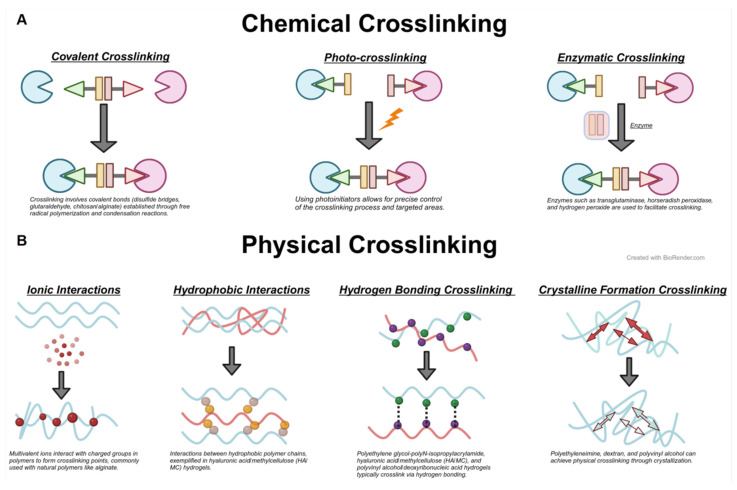
Schematic Diagram of Hydrogel Cross-Linking Methods. (**A**) Classification and principles of chemical cross-linking in hydrogels. (**B**) Classification and principles of physical cross-linking in hydrogels. The small balls represent different atoms or ions involved in cross-linking. The gray arrows represent crosslinking reactions. Red and green arrows represent electrostatic forces.

**Figure 5 biomedicines-12-00923-f005:**
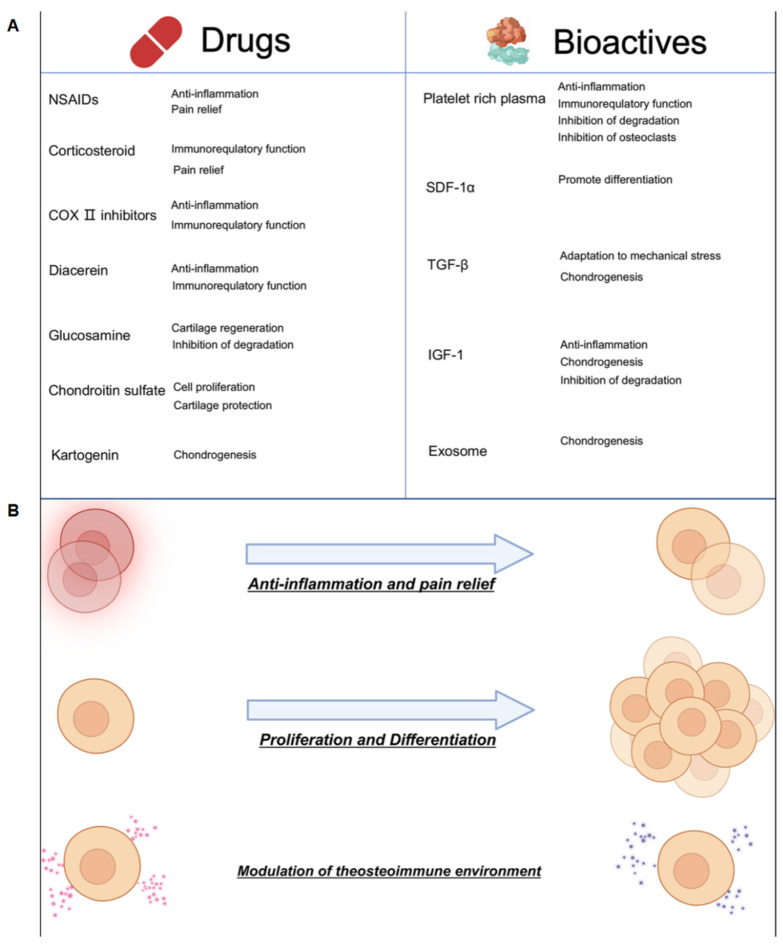
(**A**) Drugs and bioactive substances in hydrogel delivery systems used for treating OA. (**B**) Principles and mechanisms of hydrogel treatment for OA.

**Figure 6 biomedicines-12-00923-f006:**
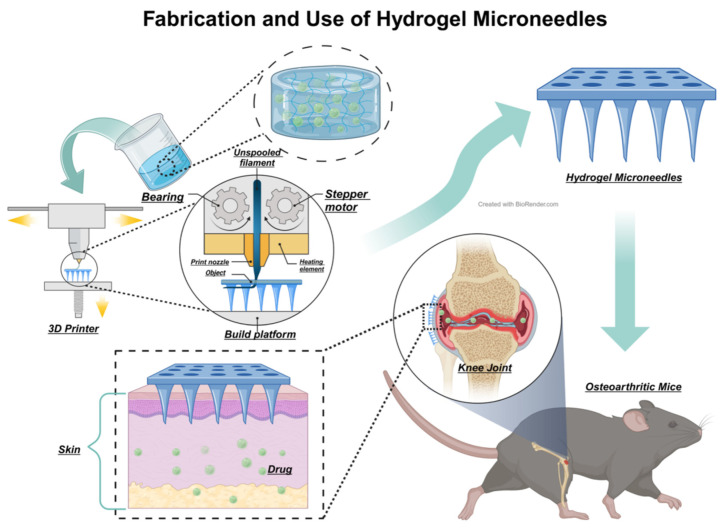
A possible and feasible method of creation and therapeutic process of hydrogel microneedles for treating OA can be described as follows: After successful loading of drugs or bioactive substances and subsequent crosslinking, hydrogel is formed into a microneedle array using 3D printing technology. These microneedles, exceedingly small in size, are designed to penetrate the stratum corneum, the outermost layer of the skin, without affecting underlying nerves. Specifically engineered for targeted joint areas, these hydrogel microneedles, upon penetrating the skin, facilitate the release of the encapsulated medication into the body, thereby providing more precise and localized treatment for OA [126].

**Figure 7 biomedicines-12-00923-f007:**
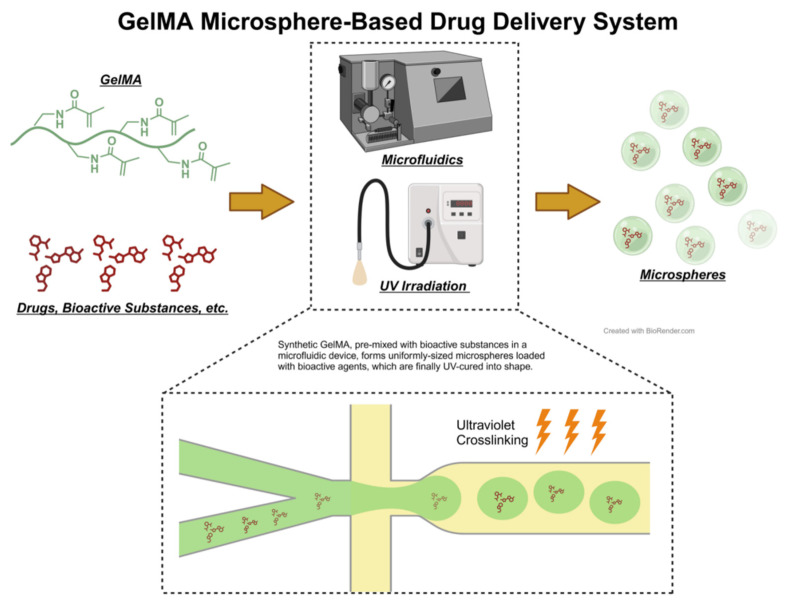
Illustration of the Principles of Microfluidic Technology and Photocuring Molding Technique Using GelMA Hydrogel Microspheres as an Example. GelMA, combined with the desired drugs or bioactive substances, passes through the micro-orifices in the microfluidic device to form uniformly textured microspheres loaded with the drug. These are then solidified and molded under ultraviolet light radiation.

**Figure 8 biomedicines-12-00923-f008:**
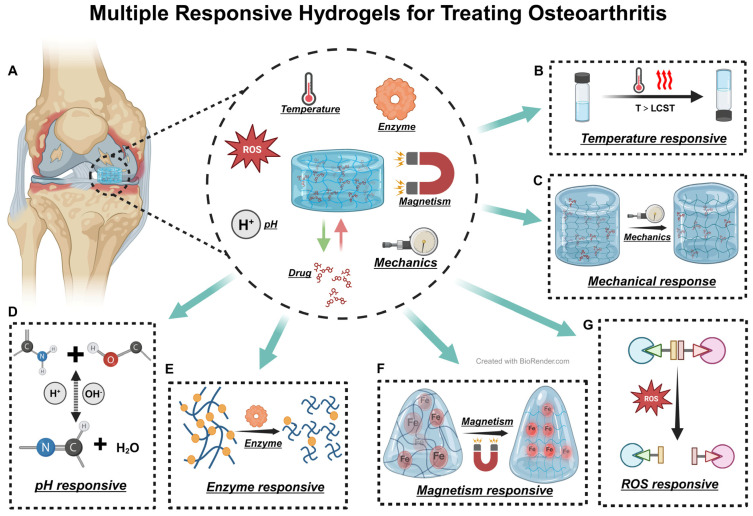
Multiple Responsive Hydrogels for Treating OA. (**A**) Diagrammatic representation of OA. (**B**–**G**) Illustrations depicting the principles of temperature-responsive, mechanical-responsive, pH-responsive, enzyme-responsive, magnetic-responsive, and ROS-responsive hydrogels, respectively. Black arrows represent responsive reactions.

**Figure 9 biomedicines-12-00923-f009:**
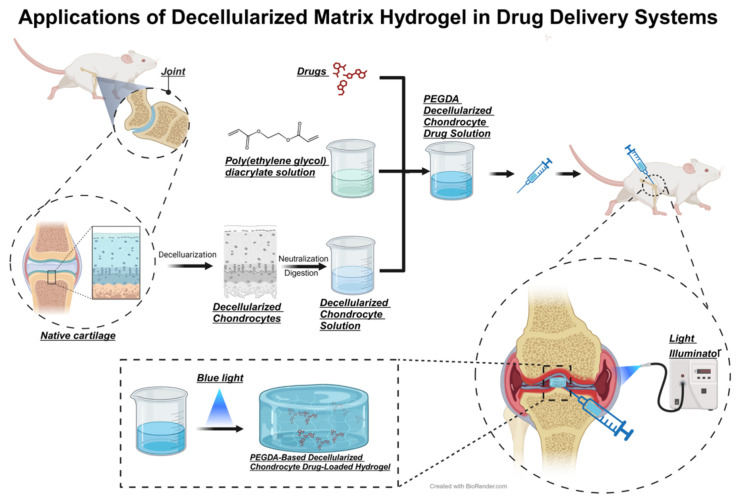
Preparation of Decellularized Matrix Hydrogel Derived from Rat Chondrocytes and Its Application in OA Treatment. Rat cartilage undergoes decellularization to form a sol containing the extracellular matrix of chondrocytes. This is then cross-linked with PEGDA to form an injectable, photosensitive decellularized matrix hydrogel drug delivery system, which gels inside the body under blue light radiation.

**Figure 10 biomedicines-12-00923-f010:**
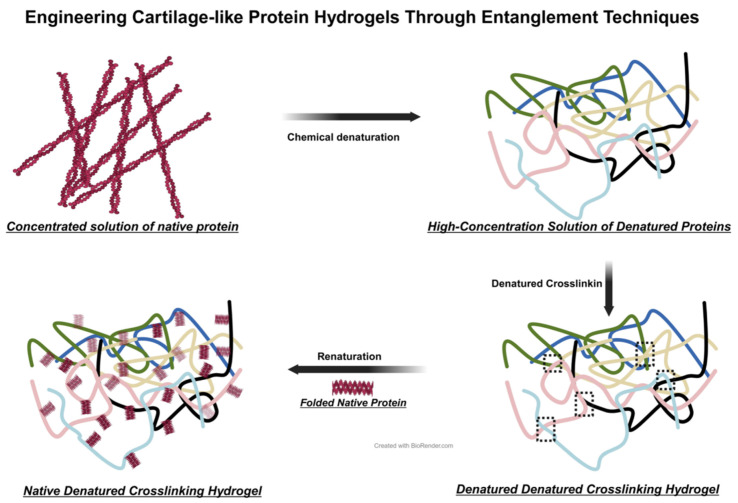
Schematic Diagram of Entangled Cross-Linked Protein Hydrogels. The gelation process of entangled protein hydrogels includes four stages: concentration of solution, chemical denaturation, denaturation cross-linking, and renaturation folding.

**Figure 11 biomedicines-12-00923-f011:**
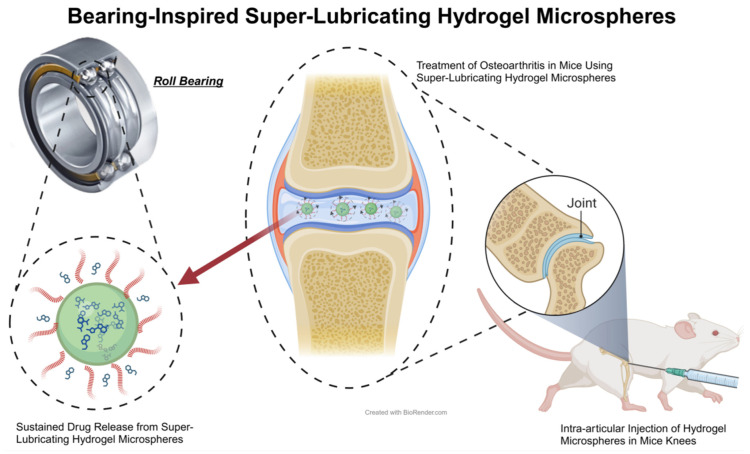
Schematic Diagram of Superlubricating Hydrogels. The diagram illustrates the design of ball bearing-inspired superlubricated microsphere, which synergistically treats OA in rats. The advent of this hydrogel marks a significant milestone in the treatment of OA, through enhanced hydration lubrication and sustained drug release [144].

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
