# Peer review of "Applications of Hydrogels in Osteoarthritis Treatment"

_biomedicines, 2024, doi:10.3390/biomedicines12040923_

Round 1

Reviewer 1 Report

Comments and Suggestions for Authors

Reviewer’s comments:

The review manuscript entitled ‘Application of Hydrogels in Osteoarthritis Treatment’ has been peer-reviewed. In the present work, the authors have highlighted recent advancements in the application of innovative hydrogel materials to treat osteoarthritis. The review encompasses technological progress and a comprehensive understanding of hydrogel-based OA therapies. I recommend the manuscript be accepted in its current form. 

Comments on the Quality of English Language

Minor editing of English may be required.

Author Response

Thank you very much for taking the time to review this manuscript. The reviewer comments are laid out below in italicized font and specific concerns have been numbered. Our response is given in normal font and changes/additions to the manuscript are given in red text.

Reviewer 2 Report

Comments and Suggestions for Authors

The review focuses on “Application of hydrogel in osteoarthritis treatment.” In this review article, the author discussed the physicochemical properties and the applications of various hydrogels (natural and synthetic) in the treatment of osteoarthritis. The content of the review and the way of the organization were good. However, the novelty of the content is highly questionable.

1.      How this review has differed from the following recent review articles.

Ø  Duan, WL., Zhang, LN., Bohara, R. et al. Adhesive hydrogels in osteoarthritis: from design to application. Military Med Res 10, 4 (2023). https://doi.org/10.1186/s40779-022-00439-3.

Ø  Wang S, Qiu Y, Qu L, Wang Q, Zhou Q. Hydrogels for Treatment of Different Degrees of Osteoarthritis. Front Bioeng Biotechnol. 2022 Jun 6;10:858656. doi: 10.3389/fbioe.2022.858656.

2.      The originality of the review should be highlighted by comparing previously published articles.

3.      The conclusion of the article is missing.

Comments on the Quality of English Language

Minor editing of English language required

Author Response

(The authors gave the same response as above.)

Reviewer 3 Report

Comments and Suggestions for Authors

This review critically examines recent advancements in multifunctional hydrogels, focusing on their emerging applications in osteoarthritis (OA) therapy. It highlights a shift towards synthetic and composite hydrogels, which offer superior mechanical properties and biodegradability compared to traditional natural materials. Novel applications such as injectable hydrogels, microneedle technology, and responsive hydrogels are explored, showcasing their role in targeted and efficient therapeutic delivery for OA treatment. Additionally, the review discusses hydrogel materials, including natural and synthetic super-lubricating hydrogels, which hold promise in reducing joint friction and inflammation. The integration of bioactive molecules within hydrogels to enhance therapeutic efficacy is also addressed. The review anticipates continued technological advancements and a deeper understanding of hydrogel-based OA therapies, emphasizing their potential for tailored, minimally invasive treatments and their significant contribution to advancing biomaterial science for OA management.

However, the manuscript could benefit from additional characterizations and data to enhance the results and discussion sections. Some of the points with the current manuscript are outlined below, and it is suggested that these be addressed in a revision. Therefore, the reviewer recommends that the paper be reconsidered following these revisions.

The reviewer has the following comments:  

Abstract Revision: The abstract should be refined to align with a more scientific style. It is essential to prioritize the discussion of the extensive data encompassed within the manuscript. A concise summary of key findings and their implications should be highlighted prominently.

Figure 1: A schematic cartoon should be reported as scheme 1 and it should feature a unique and innovative infographic that visually represents the key concepts including advanced drug delivery systems and hydrogels and others discussed in the manuscript.

Figure 4: A complete structure of gelatin should be added to Figure 4.

Table 1: The current clinical status of the current systems for osteoarthritis should be reported.  

Section 2.1: This section is very small, there is so much literature on natural and synthetic polymers, so, this section should be expanded.

Section 4. Advanced Drug Delivery Systems: This section appears to be quite brief and would benefit from a comprehensive revision to help readers clearly understand the scientific problems addressed by this review. As a biodegradable and biocompatible protein derived from collagen, gelatin has been extensively utilized as a foundational element in biological scaffolds and drug-delivery systems within the context of precision medicine. The malleable nature of natural gelatin/GelMA, which can be readily engineered, holds substantial promise for the development of diverse delivery systems aimed at safeguarding and augmenting the efficacy of pharmaceutical agents. These endeavors are poised to bolster the safety and efficacy of a wide spectrum of pharmaceutical products. To offer a comprehensive exposition of natural polymer's multifaceted attributes and applications, it is prudent for the authors to reference recent scholarly works authored by Ramiro Manuel Velasco, EV Barrera, Mamidi, N., Fatemeh Ijadi, and Javier Villela. Consequently, they facilitate a thorough elucidation of the merits associated with natural composites as preferred biomaterials, with a particular emphasis on aspects like biocompatibility, degradation kinetics, and controlled drug release, thereby fostering a comprehensive comprehension of its OA advantages.

Mechanical Properties of Hydrogel Derivatives: The major downside of natural polymers is their poor mechanical properties and water solubility. Therefore, it is necessary to functionalize the gelatin to improve mechanical properties and prolong degradation. Thus, mechanical properties are of utmost importance in OA applications. There should be a Table comparing these properties.

Section 6. Discussion and Outlook: This section is not clear and too long. It should be considered concise.

Prospective: A separate section of Prospective should be reported.

Commercial Hydrogel Scaffolds: To make the manuscript more comprehensive, commercial hydrogel scaffolds used in drug delivery systems can be discussed before concluding the manuscript. Thus, a separate section should be added on commercial gelatin scaffolds and the clinical status of different gelatin scaffolds.

Conclusions: Considering the updated data, it is advisable to revise the conclusions to incorporate a more substantial quantitative dataset.

Author Response

Thank you very much for taking the time to review this manuscript. We sincerely thank the editor and all reviewers for their valuable comments, which have greatly helped us to improve the quality of our manuscript. The reviewer comments are laid out below in italicized font and specific concerns have been numbered. Our response is given in normal font and changes/additions to the manuscript are given in red text.

Round 2

Reviewer 2 Report

Comments and Suggestions for Authors

-

Author Response

Thank you very much for your careful review. We are glad for your reply!

Reviewer 3 Report

Comments and Suggestions for Authors

No more comments!

Author Response

Thank you very much for your careful review! We are glad to receive your comments!